# Climate-Related Natural Disasters: Reflections on an Agenda for Rural Health Research

**DOI:** 10.3390/ijerph20085553

**Published:** 2023-04-18

**Authors:** Ross Bailie

**Affiliations:** 1Sydney Medical School, The University of Sydney, Lismore, NSW 2480, Australia; ross.bailie@sydney.edu.au; 2Sydney School of Public Health, The University of Sydney, Lismore, NSW 2480, Australia

**Keywords:** climate change, rural health, health equity, community-based research, extreme events, health system resilience

## Abstract

The increasing frequency and severity of climate-related disasters will exacerbate the health inequities that already exist between people living in rural communities and those living in urban areas. There is a need to improve understanding of the differences in the impacts on and needs of rural communities, in order that policy, adaptation, mitigation, response and recovery efforts meet the needs of those who are most affected by flooding and who have the fewest resources to mitigate the impact and adapt to the increased flood risk. This paper is a reflection by a rural-based academic on the significance and experience of community-based flood-related research, with a discussion of the challenges and opportunities for research on rural health and climate change. From an equity perspective, there is a need for all analyses of national and regional datasets on climate and health to, wherever possible, examine the differential impacts and policy and practice implications for regional, remote and urban communities. At the same time, there is a need to build local capacity in rural communities for community-based participatory action research, and to enhance this capacity through building networks and collaborations between different researchers based in rural areas, and between rural- and urban-based researchers. We should also encourage the documentation, evaluation and sharing of experience and lessons from local and regional efforts to adapt to and mitigate the impacts of climate change on health in rural communities.

## 1. Introduction

During 2022, the Australian State of New South Wales (NSW) experienced exceptionally extensive and severe flooding. These floods followed the worst bushfire season on record in late 2019 and early 2020, and coincided with the disruption caused by the COVID-19 pandemic. One of the towns most severely affected by the 2022 floods was Lismore, a regional town of about 30,000 people in the Northern Rivers region of Northern NSW. The town is a major regional agricultural, business, health and educational service centre.

Lismore has experienced frequent previous flooding because of its location on a floodplain at the confluence of two rivers. Following severe flooding in the town in 2017, we established a community–academic partnership to undertake research into the mental health impacts of this flood event [1]. In the context of widespread flooding across rural NSW in 2022 and the prospect of increasingly severe and frequent flooding, this paper is a reflection, as a rural-based academic, on the experience of flood-related research, with a discussion of the challenges and opportunities for research on rural health and climate change.

## 2. The Health of Regional and Remote Communities and Climate-Related Disasters

In NSW, as in other parts of Australia and internationally, people living in rural locations have poorer health outcomes than their non-rural counterparts [2,3,4]. Almost 30% of people in Australia live outside our major cities, with a clear gradient towards poorer health outcomes from major cities, to inner and outer regional areas, and to remote and very remote locations [5]. Underpinning the gradient in health outcomes is a gradient in access to and quality of health care, and in increasing workforce shortages [6]; and underpinning that is a gradient in the social determinants of health—for example, education, employment and income levels [3]. Complicating this general gradient are local inequities, with many regional and remote areas having high levels of poverty and housing disadvantage, while others are relatively well-off [7].

The disparity between major cities, regional and remote areas is arguably one of the greatest axes of inequity in Australia—because of the magnitude of the disparity and because of the numbers of people involved, with almost 30% of all Australians living in regional and remote locations [5]. The categorisation of major cities, inner and outer regional areas, and remote and very remote locations follows the Australian Statistical Geography Standard (ASGS) [8]. In this paper, I use ‘rural’ to refer to regional and remote locations in general (as defined by the ASGS), noting there is no formal definition for ‘rural’ within the ASGS. While the gradients in health outcomes and the factors underpinning these, as described above, are recognised in Australia and internationally, there is a need for better analysis and reporting of the differences in health outcomes and the factors impacting health [4].

Natural disasters, including climate-related events, expose and exacerbate the inequities that exist in our society and in local communities. Those with fewer resources are impacted more severely by these physical events; and because they have fewer resources they are less able to respond to, manage and recover from the events [9,10]. The impacts on health of these events are greater for those who are least able to cope because of socio-economic circumstances, weaknesses in social structures, or health or other conditions that make them more vulnerable [10]. The impacts are most severe for the groups of people who already face greater health challenges, whose health tends to be relatively poor, and who have greater difficulty accessing health and other services that they need [11,12,13].

Thus, the rural context and climate-related disasters compound the health challenges generally faced by groups that have a higher need for health and social support services and/or marginalised groups—an example of the classic ‘triple whammy’.

As for previous floods, the two major floods that occurred in February and March 2022 in the Northern Rivers have had complex, multifaceted and interrelated impacts for communities across the region [14]. The impacts have been different for different locations and different rural communities, partly because of pre-existing social, economic and geographic differences in these communities, and partly because of differences in the weather events between local areas (where the rain falls, how much rain over what period, local impacts and impacts downstream from catchments). Local responses and capacity to respond also resulted in differences in impact between communities.

The events led to loss of life, severe injury, fear, loss of property and of livelihoods, grief and distress in the immediate and longer term. The impacts on human health and health services, and the response and recovery of health services, have been only a part of the wide range of impacts and of response and recovery efforts. During these events and their immediate aftermath, health and health services were not at the forefront of most people’s minds. Instead, the focus was on survival and on meeting basic needs—protecting and caring for family, friends and property; finding shelter, food and a dry and warm space to rest and recover; and providing immediate psychological support. Alongside these immediate and ongoing needs, health issues have become of increasing relative importance after the immediate response phase, and these health issues continue to evolve over time. This change is occurring alongside continuing demand on resources to support recovery in other sectors, for example in public infrastructure such as roads and bridges, as well as in housing, schools, business and industry.

The health impacts, especially the mental health impacts, of such events can last for years, particularly in the context of recurrent events. The recurrence of events exacerbates the sense of fatigue experienced by many. There is an interplay and shifting balance between the sense of fatigue and hopelessness on one hand, and the sense of determination, resilience and drive for recovery on the other.

The impacts on health services and the health workforce have been profound. There has been destruction or severe damage to primary health care and other health-related service infrastructure, sometimes with irrecoverable loss of health and other records; people have not been able to get to work because of flooded roads, leaving health services understaffed; people have not been able to get access to regular medication, or to attend routine appointments; and there has been an increase in Emergency Department presentations with flood-related injuries and infections. The loss of housing and accommodation has meant that some people cannot be discharged from hospital because they have nowhere to live, resulting in the overcrowding of hospitals, additional pressure on staff and an inability to admit patients in need; and there has been difficulty attracting junior doctors, registrars and other new staff because they cannot get accommodation in the area, exacerbating existing rural health workforce shortages.

## 3. Community-Based and Co-Designed Research Following the Northern Rivers 2017 Floods

Consistent with international experience [15,16,17,18], our local community-based and co-designed research had substantial direct and collateral benefits following the 2017 Northern Rivers floods. The stakeholder consultation forums established for research became forums for more general local stakeholder networking and information sharing. The strong community response and engagement meant we were able to obtain data from a range of marginalised groups. Our research resulted in significant presentations in community, service and academic forums, research publications, submissions to local government and royal commissions, and parliamentary enquiries. The knowledge and experience gained through this collaborative research has been valuable for informing response and recovery planning for the 2022 Northern Rivers floods and for flood events affecting other communities over recent years.

The methods and findings of our research following the 2017 floods have been previously published [1,11,12,13,19,20,21,22]. In the context of more recent extensive and widespread flooding, and the probability of more severe and frequent floods, our research has relevance to policy, practice and further research. This perspective paper does not review the findings of this research or their significance, other than to briefly highlight three key findings and their implications: (a) that the length of displacement from homes was strongly associated with poorer mental health outcomes [11,12,13]—it is thus important to get people back into their own homes, or at least into stable accommodation, as soon as possible; (b) that community connectedness, or social capital, was protective for mental health—thus the evidence on how to promote community connectedness/social capital is important for informing recovery and rebuilding programs [22]; and (c) that there are high-risk groups that have specific needs—it is thus vital to include groups who are at high risk of negative impacts in planning response and recovery efforts, such as Aboriginal and Torres Strait Islander peoples; people with a disability and their carers; young people and older people; those in poorer socio-economic circumstances; and people who are homeless or in transient housing [11,12,13].

Our research work was valuable for understanding the impacts of and supporting recovery from the 2017 flood, and for informing responses to the 2022 flood. We are continuing to publish follow-up research and explore new projects. However, our experience of the flood events as rurally based researchers in the Northern Rivers has highlighted some general challenges for research on the health impacts of climate change and related disasters in the rural context, including in relation to the co-design of research, the priorities for research focus, appropriate research methods and translation to policy and practice. This experience resonates with studies and consensus statements on rural health research in Australia more generally, including the need to build rurally based research capacity and the importance of research networks [23,24,25]. My perspectives on key challenges are outlined in Box 1.

Box 1Challenges of community-based rural health and weather-related disaster research.The ability of researchers to engage at local community level depends on local presence and pre-existing relationships.This is especially important in determining the ability
to reach marginalised groups (such as people with a disability, carers, people
who are homeless, Aboriginal and Torres Strait Islander peoples, people with
a low socio-economic status) and groups that may be more severely impacted
because of the threats to livelihood (e.g., farmers, small business owners)Reaching different priority groups may require special
skills, experience, connections, relationships, and resources.There are dilemmas about targeting research efforts to
reach specific groups rather than aiming to gain broader understanding.Research grant funding timelines and processes mean it is
difficult to get timely access to funding and resourcing for research on the
impact of sudden events that occur at uncertain intervals, and across
different locations.There are important differences between rural communities
that will shape research priorities and possibilities and raise questions
about the generalisability of findings.There are complex issues related to study design,
sampling, ‘recruitment’ and engagement, and the generalisability of findings.
It takes time to work through these issues, especially in the context of
co-design in stressed and overstretched communities.There are a range of considerations regarding the
availability and interests of local rural researchers and metropolitan-based
researchers, and their relative roles in the aftermath of climate-related
disasters. For example:
○
Local researchers may have local
knowledge and connections, but limited capacity to respond because of their
low numbers and scattered locations, and because they have other personal and
work commitments that are affected by the climate-related disasters and
response and recovery process;○
Local researchers may themselves be
stressed and fatigued by the events, and may have limited capacity to take on
additional work;○
Local people may feel the need to
prioritise support for response and recovery efforts, and to carry out more
general volunteer work that addresses immediate needs rather than conduct research;○
Local people may be more concerned
about local community fatigue and stress for community members, and may feel
it is not appropriate for people to be approached regarding disaster-related research
in the context of other pressing demands and stresses;○
Metropolitan-based researchers may
have greater capacity but less ability to get local engagement, since they may
have less insight into local priorities, may have their own agendas and do not
necessarily have the connections to gain local insights;○
Metropolitan-based researchers may
press ahead with their research as they may see this as a way they can
contribute, but without the relevant skills and attention to local community
circumstances, their research may do more harm than good, their goodwill and
enthusiasm can place additional demands on local communities, and the lack of
communication and coordination can result in misplaced effort and duplication
and the inefficient use of resources.


## 4. Challenges in Defining a Research Agenda for Climate-Related Rural Health Research

The reflections on researching the health effects of the floods in the Northern Rivers have also identified important challenges and questions in defining an agenda for further flood-related rural health research. These reflections are relevant to other climate-related health research because rural communities are in many respects at the frontline of the battle to manage and respond to the impacts on health of climate-related events. Globally and in Australia, the range, severity and complexity of the impacts of climate-related events are greater for rural communities than for major cities because of the lower response capacity [9,10,26] and because of greater exposure (as evident in historic and projected exposure to climate-related disasters such as flooding, sea level rise, drought, heatwaves, water scarcity and wildfires [9,26,27,28,29]). Vulnerability to climate change is multifaceted [26] and the impacts can be fast-moving and dramatic (e.g., severe storms, floods, bushfires), intense and recurrent (extreme heat); slow-moving and relentless (e.g., droughts, sea level rise, with the potential loss of rural communities whose livelihoods are based on agricultural land on coastal floodplains); indirect (e.g., changes in the geographic range and incidence of mosquito-borne disease, in agricultural conditions, economies, mining, with associated impacts on health, livelihoods and wellbeing); and immediate, evolving or long term. While there are examples of research that explores rural–metropolitan differentials in exposure, vulnerability or health outcomes related to climate change (see for example [26,27]), research on these differentials has not received the attention that the inequities and complexities demand.

The research agenda is therefore very large and complex: how should we determine priorities for research on climate and health in rural contexts? Who should be conducting this research, how should we build the capacity for rural research, and how can we make the most effective use of local research capacity and capability? What methods are the most appropriate: quantitative, qualitative or mixed methods, or participatory action research? Of course, this depends on the research question, and we probably need capacity across the spectrum of methods. I propose three strategies for climate-related research to contribute more effectively to regional and remote health (Box 2).

Box 2Strategies for climate-related research to contribute more effectively to regional and remote health.
*
**Examine differential impacts across major cities,
regional and remote communities**
*
Possibly the greatest contribution could be through all
researchers considering the potential for differential
impacts between major cities and regional and remote communities.
Researchers tend to have relatively specific interests, and often do not
characterise their work as urban, regional or remote. However, rurality has
an important influence on the impacts of climate-related events, and the
disparities in socio-economic circumstances, health status and access to
services should be important considerations in policy and planning for
adaptation, mitigation, response and recovery. Wherever possible, in the
contexts of their specific interests, researchers should examine the
differences in impacts and the different implications for policy and practice
in relation to health equity and implementation [30]. In relation to
climate and health, this includes examining differential impacts and policy
and practice implications for major cities, regional and remote locations in
general, for specific types of communities and for specific community groups.

*
**Build capacity to conduct high-impact research in rural
communities**
*
It is vital to provide support for research to be conducted
in regional and remote locations as well as in major cities. Building
research capacity in rural communities is a high priority if we are to carry
out high-quality rural health research. There are substantial benefits to local
in-depth and context-specific research, and we should continue to conduct this.
However, we should also be improving our efforts to carry out local research
in a way that has a greater potential for large-scale impacts in regional and
remote Australia, or internationally.
*
**Build rural health research networks that include links
to urban-based researchers**
*
One approach may be to develop networked research
capacity that is funded to prepare for and respond to carrying out research
at multiple local levels as the need arises. The networks could be across
rural locations as well as between metropolitan and rural locations. The
networks could facilitate the sharing of tools, resources and experience with
researchers studying similar events in other locations. There may be
potential for the pooling of data and/or comparing experience and findings
across multiple local areas.

## 5. Conclusions

The proposed strategies have implications for many stakeholders. Rural- and urban-based climate and health researchers should consider questions of equity between urban, regional and remote communities (and sub-groups within these communities) in relation to exposure to climate-related events and the ability to respond. Researchers should engage in co-design with community members, service providers and policy makers to ensure problems are understood from various perspectives and appropriate solutions developed and evaluated. Researchers should aim to develop multistakeholder networks and collaborations with the aim of building capacity and maximising understanding of generalisable/transferable messages and of factors that may be important to local context. Community members, service providers and policy makers should be proactive in engaging with researchers to ensure their perspectives and needs are understood and addressed.

Noting the significance of rural health inequities, the underfunding of rural health research in general [31] and the huge scale of the health-related risks of climate change, universities and research institutes should increase their commitment to the development of climate-related rural health research and rural health research capacity, including through investment, advocacy and supporting collaboration; research funding bodies should target funding for climate change and rural health inequities; and government agencies responsible for health and welfare data should report on climate-related rural health inequities.

I hope that this reflective paper will stimulate interest and ideas, and will contribute to the debate on rural health and climate change. This is a very significant and challenging area, and one that has enormous potential for the development of research that can make a difference in our society.

## Data Availability

No new data were created for this paper.

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
