# Peer review of "Climate-Related Natural Disasters: Reflections on an Agenda for Rural Health Research"

_ijerph, 2023, doi:10.3390/ijerph20085553_

Round 1

Reviewer 1 Report

The paper is well-introduced and covers a relevant and timely topic. The introduction does an excellent job of setting the stage for why a research agenda is important and uses good a strong case example. I would suggest the author be clearer when describing the specific case example, perhaps separating it out via a sub-heading.

Despite the above, the structure of the paper is not suitable for this journal.

Specifically, beyond line 89, much of the paper is written as a mix of dashes, numbered paragraphs, or bullet points. The content itself is good, but it would be more appropriate to be written more formally, with short highlights of the framework appearing in several well-placed tables. This would allow readers to quickly see the most important elements and refer back to the main body for detail. 

Author Response

Reviewer's comment - 1. Although the title of the paper is too interesting and relevant, the contents seems to be just observations and conclusion.

Response: Thank you for your positive comments on the introduction to the paper and on the relevance and timeliness of the topic, also for your positive comment on the content of the paper and for your suggestions for improving presentation.

Reviewer's comment - 2. Despite the above, the structure of the paper is not suitable for this journal. Specifically, beyond line 89, much of the paper ...

Response: I have revised the structure of the paper to include a revised a subheading to more clearly refer to our work following the Northern Rivers 2017 flood. 

I have also removed the ‘dashes’ and presented the text as paragraphs. I have moved some text to two boxes/tables to enable readers to discern key content more readily.

Reviewer 2 Report

Although the title of the paper is too interesting and relevant, the contents seems to be just observations and conclusion. 

What percentage of people are rural and what is the impact on other areas?

Author Response

Reviewer's comment - 1. Although the title of the paper is too interesting and relevant, the contents seems to be just observations and conclusion.

Thank you for the positive comment on the interest and relevance of the paper. As stated in the abstract and in the second paragraph of the introduction, this paper is a reflection as a rural-based academic on the experience of flood-related research with discussion of challenges and opportunities for research on rural health and climate change.

Reviewer's comment - 2. What percentage of people are rural and what is the impact on other areas?

Thank you for the relevant questions. Almost 30% of the Australian population live outside our major cities. I have added reference to this in line 42 of the revised paper. While the gradients in health outcomes and the factors underpinning these are recognized in Australia and internationally, there is a need for better analysis and reporting of the differences in health outcomes and factors impacting health, including climate change. I have added text and a reference to support this in lines 57 to 58 of the revised paper.

Reviewer 3 Report

This perspective presents a single-author reflection, as a rural-based academic, “on the significance and experience of community-based-flood-related research with discussion of challenges and opportunities for research on rural health and climate change.”   The perspective is presented from an equity point of view, based on rural communities being disadvantaged on many fronts compared with their rural counterparts.  The author calls for research to examine the differential impacts and on rural and urban communities, to build local capacity in rural communities to undertake community-based participatory action research, and to build networks and collaborations between rurally-based researchers and between rural and urban based researchers.

The author addresses an area of great importance given the more frequent occurrence of extreme weather events (i.e. the effects of climate change) in recent times.   To be commended is that the author has clearly led research in this space as demonstrated through references 1 – 5 of this Perspective.   These five references also support the author’s collaboration with other researchers to undertake research in this space.   In view of these outputs from research collaborations, it is puzzling why this is a single-author Perspective and what the gap is in collaboration and establishing networks to undertake further research in this space.   A quick scan of full-texts of references 1 – 5 suggests that the authors have done some important work highlighting the health effects (largely mental health and wellbeing) experienced by disadvantaged groups during a major flooding event.  The author and collaborators should be well-placed to advance their preliminary work through competitive funding schemes.  Climate change appears to be a priority for many competitive funding schemes despite the fact that population health research is often not prioritised for Australian Government funding schemes.  

Nevertheless, such perspectives are necessary to bring to the attention of decision makers the impacts of natural disasters on already disadvantaged rural communities.  There is never too much written about this important issue to help bring it to the forefront of policy makers’ priority lists.

I make the following suggestions for the author to consider, not as criticism of the Perspective but more in terms of improving the likelihood of it being more impactful in getting the necessary attention.

Make the Perspective more concise.  The current version is somewhat lengthy.

2.   Include more of an evidence-based to support the claims.  While supportive evidence has been included, there are research publications that are very relevant but not been referenced.  For example, a quick and dirty search for literature identified the following recent publication that examined the very topic addressed in this Perspective, albeit not limited to floods.  How do the findings from this published paper align with the reflections of the author? There are other potentially relevant references in this publication.

Wang, S., Zhang, M., Huang, X. et al. Urban–rural disparity of social vulnerability to natural hazards in Australia. Sci Rep 12, 13665 (2022). https://doi.org/10.1038/s41598-022-17878-6

 3.       Draw more attention to the findings from the five co-authored papers that the author refers to (references 1 – 5).  The findings are important and would provide a good foundation for addressing some of the gaps that the author includes in this perspective article.

A question to the author:  Many of the Group of Eight Universities have rural campuses with researchers funded through RHMT and other funding sources (e.g. UDRH and other rural research centres).   Why is it that academics located at these well-established and well-funded campuses not been able to address these needs?   What is preventing these rurally-based academics establishing networks and collaborations?   Do we need to understand factors inhibiting such collaborations before drawing attention to particular areas of need?

Author Response

Thank you for the positive comments and for the suggestions on improving the paper. In response to the question about this as a single author perspective in the context of the collaborative research and research networks: I confirm that while this perspective is a personal one, the ideas have been stimulated through discussions and work with a wide range of people. I have added an acknowledgment to this effect.

Reviewer's comment - 1.  Make the paper more concise ...

Response: I have revised the structure of the paper to enhance flow, moved text to boxes/tables as suggested by one of the other reviewers, and edited the text to make the paper more concise. However, responding to the reviewers’ comments has also resulted in additional text.

Reviewer's comment - 2.   Include more of an evidence-based to support the claims ...

Response: Thank you for this comment and the suggested reference. I have revised the text in lines 146 to 155 in the new version of the paper to address this comment. I have included a number of additional references and have reordered the references in line with the revised structure of the paper. It is beyond the intended scope of the paper to provide a review of the research on metro-rural differentials in exposure, vulnerability or health outcomes related to climate change, but note that we have done some preliminary work on a scoping review on this topic.

Reviewer's comment - 3. Draw more attention to the findings from the five co-authored papers ...

Response: Thanks for the comment on the importance of the findings of our research and the suggestion for drawing more attention to these findings. I have added other references to our research and added text in the paper to encourage readers to consider the implications of our research on policy, practice and further research (see lines 114-117 of the revised paper).

Reviewer's question to the author:  Many of the Group of Eight Universities have rural campuses ...

Response: Thanks for this question. I have been connected to the RHMT funded rural health campuses for several years through my previous lead role in one of these sites. In this role I have encouraged other sites to engage in networks and collaborations on climate change and health, with limited success. There is published research on the challenges of rural health research in Australia in general and this research highlights the lack of rural research capacity as a major limiting factor (see new references on this in the revised version of the paper). Other considerations are that a) the research focus of the RHMT sites is defined by the RHMT contracts and while these contracts do not preclude research on climate change, they do prioritise other areas of research; b) the research focus in RHMT sites is to some extent dictated by the interests and experience of the small number of researchers in these sites and it appears that to date relatively few of these researchers have been active in climate related research.

Round 2

Reviewer 1 Report

This a much-improved version of the paper and the results are now presented in a clear format. I think that the simple addition of boxes makes the results very useful as a summary of challenges for community-based researchers.

My only additional suggestion is that the Conclusion could be expanded as it seems rushed. Perhaps something could be said about who could implement some of the changes suggested, such as building rural health research networks or building local capacity. (Both are issues seen elsewhere in the literature, so having concrete suggestions would improve the strength of the article.)

Author Response

I have expanded the conclusion along the lines suggested. Thank you for this advice.